# On the Computational Efficiency of Adapting Transformer Models via Adversarial Training

## Abstract

Pretraining Transformer-based language models followed by adapting the pre-trained models to a downstream task is an effective transfer mechanism in NLP. While it is well-known that the pretraining stage is computationally expensive, even the adaptation starts to become time-consuming for many downstream tasks as Transformers grow in size rapidly. Prior work focuses on reducing the pretraining wall-clock time via increasing the batch size to obtain higher training throughput on multiple processors. However, few studies have explored how such a scheme may benefit the adaptation phase. On the other hand, adversarial training has shown improved generalization for adapting Transformer models on many NLP tasks, but it is often treated as a separate line of research, where its effectiveness under the large-batch regime is not well understood. In this paper, we show that adversarial training obtains promising model accuracy even with a considerably larger batch size. However, the computational complexity associated with this approach, due to the high cost of generating adversaries, prevents it from reducing adaptation costs with an increasing number of processors. As such, we systematically study adversarial large-batch optimization for adapting transformers from a computational complexity perspective. Our investigation yields efficient and practical algorithms for adapting transformer models. We show in experiments that our proposed method attains up to $9.8\times$ adaptation speedups over the baseline on $\text{BERT}_{base}$ and $\text{RoBERTa}_{large}$, while achieving comparable and sometimes higher accuracy than the state-of-the-art large-batch optimization methods.

## 1 Introduction

In the past few years, we have witnessed the success of transformer models [2], such as BERT [9], RoBERTa [30], T5 [35], and GPT-3 [3]. These models are trained on massive open-domain data and subsequently adapted to various downstream tasks, which have led to accuracy breakthroughs in many NLP applications[44]. Despite their remarkable performance in accuracy, training these models is extremely time-consuming given their huge model sizes, ranging from a few hundred million parameters to over billions of parameters. As a result, optimizations for faster training speed with high accuracy are the focus of a highly active research area and have a clear, practical impact.

To accelerate the training speed of large models, one of the most popular approaches is to leverage distributed training, where a mini-batch is partitioned across multiple processors (e.g., GPUs) to compute gradients locally in parallel and then aggregate the local updates [27, 30, 20, 39, 40, 37]. Under such a paradigm, increasing the batch size clearly has the benefit of improved training throughput per iteration. However, increasing the batch size has a non-trivial impact on model convergence and generation in practice. Early studies observed that increasing the batch sizes often leads to slow convergence and/or poor generation under the same training iteration budget [25]. To close the accuracy gap from large batch optimizations, prior works proposed to either increase the

number of training iterations, which limits the performance benefits of large-batch optimizations [17] or variants of adaptive optimizers such as LARS [51] and LAMB [52]. It has been empirically observed that LAMB [52] is able to speed up BERT pre-training by using considerably larger batch sizes on massive GPUs. Despite showing promising results, prior work primarily focuses on large-batch optimizations for accelerating pre-training Transformers. However, as the size of Transformers increases rapidly, reducing the training overhead at the adaption stage starts to become more prominent, e.g., with the active research that has been pushing the training time of BERT models to only a few hours or less than one hour [52, 57, 1], it takes tens of hours to fine-tune these models on MNLI [30]. Furthermore, since the adaptation of these large transformer models has been used by major players in the industry, many model scientists have to perform adaptation more frequently than pre-training the Transformers. As a result, the excessive long adaptation time hinders the turnaround time, and the aggregated training cost for adaptation is also quite high.

We aim to accelerate the adaption of pre-trained Transformer models. For this purpose, we introduce ScaLA, a method that achieves similar model adaptation quality but with significantly shorter optimization time. Especially, the contributions of our paper consist of (1) We look into projected gradient descent based adversarial training, which has shown promising accuracy results in fine-tuning Transformer models. We find that adversarial training still leads to improved generalization under the large-batch regime, which we denote as adversarial large-batch optimization. (2) Adversarial large-batch optimization helps improve generalization but makes each individual processor slower, making it difficult to actually reduce training time even with a large number of processors. As such, we perform a systematic study of how different training strategies of adversarial large batch optimization affect the computational efficiency and generalization for adapting Transformers. We find that many computations in adversarial training are redundant and only have a small impact on the final model accuracy. (3) Based on our studies, we present a novel algorithm ScaLA that injects lightweight adversaries into large batch optimization to speed up the adaptation of pre-trained transformer networks. (4) We theoretically quantify the convergence rate of adversarial large-batch optimization using techniques for analyzing non-convex saddle-point problems. (5) We conduct extensive evaluation, and our results show that ScaLA accelerates the adaptation of pre-trained Transformer-networks by up to 9.8 times over the baseline on BERT [9] and RoBERTa [30] over a wide range of natural language understanding (NLU) tasks. We conduct ablation studies to assess the impact of our approach on both generalization and computational efficiency under various conditions.

## 2   Background and Related Work

Despite the great success of pre-trained transformer networks such as BERT [9], a big challenge, in general, comes from the training efficiency – even with self-attention and parallelizable recurrence [43], and high-performance hardware [24], training transformer networks can still take a significant amount of time. One effective approach to reducing training time is through data parallelism [9, 30, 40], which motivates studies on large-batch stochastic non-convex optimizations for transformer networks [52]. These studies have raised concerns with respect to its convergence, generalizability, and training stability by observing that training with a large batch could be difficult [25, 17, 33]. Different from prior works, which mostly focus on reducing the pre-training time [52, 56, 12, 6], this work shows an effective approach to accelerate the adaptation of pre-trained models while preserving the accuracy of downstream tasks.

There has also been an increasing interests in developing efficient adaptation methods of pre-trained Transformer models [18, 46, 19, 16]. For example, [18] inserts small modules called adapters to each layer of the pre-trained model, and only the adapters are trained during adaptation. [19] adds low-rank matrices to approximate parameter updates. [34] shows that it is possible to quickly adapt to new tasks by collectively learning knowledge from multiple tasks. These methods have achieved comparable performance to standard fine-tuning on different sets of tasks. However, their focus is on reducing memory consumption of adaptation by reducing the trainable parameters needed per task. Unlike these methods, which still incur full forward/backward computation cost during adaptation, we investigate how to accelerate the adaptation speed through adversarial large-batch optimization.

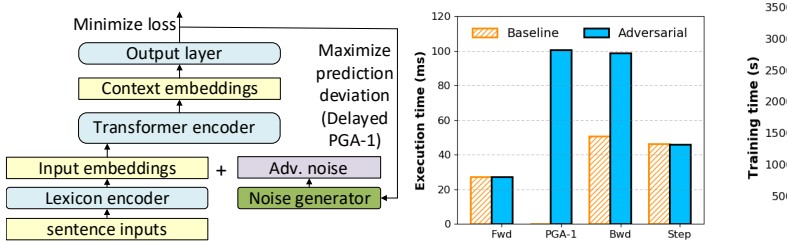 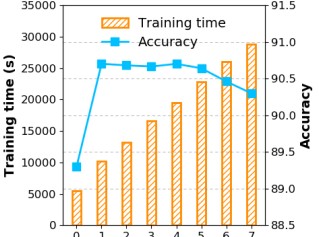

Figure 1: The architecture of the proposed method.

Figure 2: Time breakdown without and with PGA-1.

Figure 3: Impact of perturbation steps.

# 3 The Proposed Method

Motivated by the challenges in accelerating the adaptation, in this section, we present a principled large-batch optimization method via lightweight adversarial noise for improved adaptation speed while maintaining the quality of the solutions as measured by task-appropriate accuracy metrics.

## 3.1 Adversarial Training Preliminaries

Adversarial training has been proposed and studied extensively in the computer vision literature mainly for improving the robustness against adversarial attacks [13, 31]. The key idea is to apply small perturbation to input images that maximizes the adversarial loss:

$$\min_\theta \mathbb{E}_{(x,y)\sim D}\big[\max_{\|\delta\|\leq\epsilon} l(f(x+\delta;\theta),y)\big] \tag{1}$$

While adversarial training has been successfully mitigating adversarial attacks, traditional understanding is that adversarial training could hurt generalization performance. However, there has been an increasing amount of attention paid to leverage adversarial training for improving accuracy of clean data performance [48, 58, 10]. In particular, there are some studies show that adversarial training helps improve the generalizability of language modeling [5, 45, 22, 29]. However, very few works examine how adversarial learning works helps improve the adaptation speed of pre-trained Transformer models. [58] studies adversarial training under the large-batch regime, observing improved accuracy by accumulating the gradient of the parameters from each of the ascent steps and updating the parameters only once after $K$ inner ascent steps with the accumulated gradients. However, (1) [58] still requires multiple iterations to generate adversaries and injects adversaries in the full training process, leaving the major performance bottleneck from adversaries not reduced, (2) its implementation does not support multi-GPU training. In fact, no end-to-end training time reduction or speedup is reported in [58], putting a question on how useful it is in accelerating Transformer training time.

**Basic setup.** Since language expressions are quite sensitive to individual words or clauses, where noises against those would likely generate incorrect or biased training data with wrong labels [55]. We follow prior success in applying adversarial training to NLP models [32, 58] to have an adversarial training setup by applying noises to the continuous word embeddings instead of directly to discrete words or tokens.

$$\min_{x\in\mathbb{X}} \mathbb{E}_{\xi\sim Q}[g(x,\xi)] = \min_{x\in\mathbb{X}} \max_{y\in\mathbb{Y}} \mathbb{E}_{\xi\sim Q}[\underline{f}(x,\xi) + \lambda r(x,y)] \tag{2}$$

where $g : \mathbb{X} \times \mathbb{Y} \to \mathbb{R}$ denotes the overall training objective, $r : \mathbb{X} \to \mathbb{R}$ denotes the augmented regularization and $\xi$ denotes samples drawn from $Q$ (for simplicity, we slightly abuse the notation in using $\xi$ to denote the random variable, e.g. $\mathbb{E}_\xi[g(x,\xi)]$, or its empirical realizations, e.g. $\frac{1}{K}\sum_{k=1}^K g(x,\xi_k)$ for any $K$). The overall (outer) training objective involves a minimization problem in the parameter space while being stochastic with respect to the data space. The adversarial regularization (inner) term is a deterministic maximization problem operating in the data space conditioned on a fixed parameter configuration. We emphasize that this formulation is a two-player sequential [23], not simultaneous, game wherein the goal is to optimize a transformer network that is insensitive to adversarial noise.

### 3.2 Improving Adaptation Throughput via Large-Batch Optimizations

We are interested in extending the large-batch optimization to the Transformation adaptation phase using pre-trained BERT$_{base}$ model on GLUE as an example. This part presents several studies that motivate the design of the lightweight adversarial large-batch optimization approach in Section 3.3. The detailed hardware/software setup is described in Section 4.

**Scalability analysis.** First, we carry out a scalability test by varying the number of GPUs from 1 to 32, with and without communication. Different from pre-training, the adaptation stage often employs a much smaller batch size (e.g., 16, 32) than pre-training (e.g., 4096) [9, 30].We choose a batch size 32, as suggested by most literature for BERT fine-tuning [9, 30], and we divide the samples in the mini-batch among $P=\{1,2,4,8,16,32\}$ GPUs. If the per-worker batch size (e.g., 16) is larger than the maximum admissible per-worker batch size (e.g., 8), we use local gradient accumulation [14] to avoid running out of memory. Figure 4(a) shows the scalability results. For batch size 32, the training time decreases when P increases from 1 to 4. However, it quickly plateaus and even decreases with more GPUs. We find that this is because when the batch size is small, the communication overhead dominates the total execution time (e.g.,B=32 vs. B=32 (no comm)). The communication overhead is huge, especially when there is cross-machine communication (e.g., from 16 to 32), hindering the scalability of multi-GPU training. In contrast, by increasing the batch size (e.g., to 1K), the training time keeps decreasing as the number of GPUs increases because an increased batch size reduces the number of all-reduce communications to process the same amount of data and also increases the compute resource utilization per GPU (i.e., increased computation-vs-communication ratio).

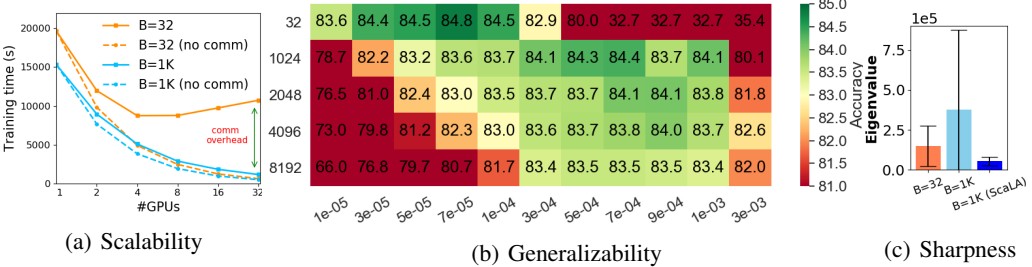

(a) Scalability          (b) Generalizability          (c) Sharpness

Figure 4: Scalability, generalizability, and curvature analysis results by adapting BERT$_{base}$ to the MNLI task.

**Generalizability analysis.** Increasing the batch size leads to accelerated per-epoch execution time due to the efficient utilization of hardware. However, how would increasing the batch size affect the generalizability in adapting transformer networks? Since prior works on batch size scaling often focus on computer vision tasks and pre-training [42, 14, 41, 52], we conduct an analysis of large-batch adaptation on pre-trained Transformers by performing a hyperparameter sweep on batch sizes {1K, 2K, 4K, 8K} and learning rates {1e-4, 3e-4,5e-4, 7e-4, 9e-4, 1e-3, 3e-3}, where the learning rate range covers both linear scaling [14] and sqrt scaling [52]. We report the validation accuracy in Figure 4(b). We make two observations: (1) the learning rate scales roughly with the square root of the increase of the mini-batch size, although the best learning rates do not always follow the sqrt rule; (2) there is a generalization gap between the small batch and large batch accuracies, and the gap becomes larger when the batch size increases. Furthermore, methods, such as LAMB [52], works well on pre-training with extremely large batch sizes ($\log_2 B = \{15, 16\}$) but do not close the generalization gap in adaptation (as shown in Section 4). These results pose the question: can we increase the batch size during adaptation in the interest of making adaptation more efficient but preserving generalization?

**Curvature analysis.** To further examine the generalization gap, we resort to the curvature analysis. Prior work [25, 54] correlate the low generalization with sharp minima (which are characterized by a positive curvature of large magnitude in the parameter space). The indication is that a sharp local minimum also reflects a higher sensitivity of the loss even within the neighborhood of training data points and can attribute to the difficulty in generalization. Their hypothesis was that a larger noise due to the higher variance in gradient estimates computed using small mini-batches, in contrast to gradient estimates computed using large mini-batches, encourages the parameter weights to exit out of the basin of sharp minima and towards flatter minima which have better generalization.

To verify this hypothesis, we quantitatively measure the steepness of loss landscape by loading the checkpoint of an adapted model and computing the curvature, i.e., properties of the second derivative of the model, with respect to its parameters, for a fixed batch of samples. Following [49], for a model $\Phi(x)$, we compute the largest eigenvalue of the model's Hessian, $L_{\max}[\nabla_x^2\Phi(x)]$, using the Hessian-vector product primitive and the power method. We use the largest eigenvalue as a measure of sharpness since the corresponding (top) eigenvector characterizes the direction of the largest change in gradient at a given point in the parameter space. From Figure 4(c), the largest eigenvalue of the model trained with a large batch (e.g., 1K) is much larger (e.g., 2.6x) than the small-batch baseline and with higher deviations (e.g., 3.9x). This result confirms that large-batch adaptation makes the loss landscape of the model more prone to ill-conditioning and less robust to perturbation, which helps explain the loss in generalization.

### 3.3  Improving the Generalization via Lightweight Adversarial Large-Batch Optimization

Our analysis indicates that although injecting adversarial noise into large-batch optimization helps improve the generalizability; it may not reduce the adaptation time because the generation of adversarial noises can take a large fraction of time. This section provides an analysis of the computational cost and then describes two approaches to reduce the time spent in generating adversarial noise, thereby further reducing the overall adaptation time.

The generation of adversarial noise requires an extra PGA inner loop that standard training does not have. Figure 2 provides the time breakdown of optimization using PGA with $\mathcal{T} = 1$ (denoted as PGA-1). PGA-1 performs the perturbation and takes approximately the same time as making three forward passes (Fwd) through the network. This is because one step of PGA requires to make one forward and backward pass (Bwd) over the entire network. The backward pass of the optimization takes roughly twice the amount of time as the standard backward step because the back-propagation is triggered twice to calculate the noise and the gradients. The time spent on the optimizer step function remains the same. In total, the optimization would slow down training by at least 2 times, even with $\mathcal{T}$=1. This motivates us to look at the effectiveness of different perturbation steps as well as the usefulness of perturbation from the initial epochs.

**One-shot perturbation.**  Prior works often do multiple gradient computation steps ($\mathcal{T} > 1$) and take several times longer training time to produce adversaries [31, 58], likely because their focus is on generalization instead of computational efficiency. Subsequently, researchers presented Curriculum Adversarial Training (CAT) [4] and Annealing-based Adversarial Training [50], which progressively increase the perturbation with various strengths, cutting the adversarial training cost while maintaining good accuracy. To investigate how CAT and similar methods affect large-scale NLP problems involving transformers, we evaluate the final accuracy and training cost of QNLI, varying the number of perturbation steps $\mathcal{T}$ and report the results in Figure 3. Interestingly, although using a large $\mathcal{T}$ helps to produce stronger noises, we find that this does not lead to improved accuracy, despite the fact that the training overhead still increases almost linearly. In fact, the best accuracy is achieved with $\mathcal{T} = 1$.

We note that the model has two components, namely, the parameter space and data space. First, unlike the minimization in the parameter space, which is stochastic, the maximization in the data space is deterministic. Second, with respect to the testing phase, the numerical convergence in the model's parameter space is of primary importance rather than the numerical convergence in the data space, i.e., the maximization is an auxiliary procedure that augments the training phase to make the parameter space "aware" of effects of the batch size across epochs. Due to these two points, at a certain epoch, for a given batch, the marginal utility of an additional PGA step is low, and we are able to get away with inexact deterministic maximization. Therefore, we apply PGA-1 in our large-batch optimization scheme, given that it produces sufficiently good solutions while being much more computationally efficient.

**Delayed perturbation injection.** Given that PGA-1 still adds an overhead factor of 2, we are motivated to further reduce the overhead of adversarial noise. In particular, we investigate how useful adversarial noises are in the whole large-batch optimization process. We conduct additional experiments to measure the final accuracy corresponding to starting from a regular fine-tuning and then enabling PGA-1 for $t \geq t_s$ where $t_s \in [T]$. Our observation is that enabling PGA-1 from the beginning does not offer much improvement in accuracy, whereas adversarial noise becomes more potent as the model begins to stabilize towards the end of training (more detailed results in

Appendix A.1). In general, at initialization, the model's parameters are relatively far from their final values and are less likely to get stuck at local minima. Therefore the adversarial noises generated in the initial training iterations are quite different from the noises towards the end of training because they would not maximize the adversarial loss in Equation 2. This hypothesis suggests that we might be able to inject adversarial noise in the later training process while still leveraging it to improve generalizability. We remark that this phenomenon has been observed by prior work on computer vision tasks [4, 15].

**Putting it together.** Combining the formulation with the above investigations, the full procedure of ScaLA is provided in Algorithm 1, whose convergence rate is characterized in Theorem 3.1.

---

**Algorithm 1**                                                                             **ScaLA**

1: **Input:** Epochs $T$, delay $t_s$, perturbation (inner) step size $\rho$, clipping radius $\omega$, regularization strength $\lambda$, (outer) learning rate $\eta$
2: **Output**: $h$-layer transformer model $\Phi$ with converged robust parameters $\overline{x} := x_T$
3: **for** $t \in [T]$ **do**
4:     **for** worker $p \in [P]$ **do**
5:         **for** mini-batch $\xi_p \sim Q$ **do**
6:             $\underline{r}(x_t) \leftarrow 0, \gamma \leftarrow \Phi(x, \xi_p)$, select $y_0$
7:             **if** $t \geq t_s$ **then**
8:                                                         $\triangleright$ Check delay condition
9:                 $y_1 \leftarrow \Pi_\omega(y_0 + \rho \nabla_y r(x_t, y))$    $\triangleright$ Generate adversarial noise with PGA-1
10:                 $\underline{r}(x_t) \leftarrow \text{KL}_{\text{sym}}(\gamma, \Phi(x_{t-1}, y_1))$
11:             **end if**
12:             $g(x_t, \xi_p) \leftarrow \underline{f}(x_{t-1}, \xi_p) + \lambda \underline{r}(x_t)$
13:             $\nabla_x g(x_t, \xi_p) \leftarrow$ Backward pass on $\Phi$
14:         **end for**
15:     **end for**
16:     $\widehat{\nabla}_x g(x_t) \leftarrow \frac{1}{B} \sum_{p=1}^{P} \nabla_x g(x_t, \xi_p)$
17:     $x_t^i \leftarrow x_{t-1}^i - \eta_t \widehat{\nabla}_x g(x_t)$
18: **end for**

---

**Theorem 3.1** (Complexity of Algorithm 1; Informal – Details in Appendix D). *Consider the problem in Equation 2. Let $t_s = 0$. Setting the outer learning rate as $\eta = O\left(1/\sqrt{T}\right)$ and scaling batch size as $b = O(T)$, for Algorithm 1, we have $\mathbb{E}\left[\|\nabla g_{1/2\alpha}(\overline{x})\|^2\right] \leq O\left(\epsilon + \kappa_\alpha/\sqrt{T}\right)$ where $\overline{x}$ is the estimator obtained from running $T$ steps of Algorithm 1 and picking $x_t$ uniformly at random for $t \in [T]$. Here, $\epsilon$ is the error due to the approximate inner maximization oracle, $\alpha$ characterizes the smoothness of $f(x, .)$, $g_{1/2\alpha}$ is the Moreau-envelope of $g$ and $\kappa_\alpha = \max_i \alpha_i / \min_i \alpha_i$.*

## 4 Evaluation

We evaluate the effectiveness of ScaLA in adapting pre-trained transformer networks over a set of NLP tasks.

**Hardware.** We conduct the evaluation using 2 NVIDIA DGX-2 nodes. Each node consists of 16 NVIDIA V100 GPUs. The nodes are connected with InfiniBand using a 648-port Mellanox MLNX-OS CS7500 switch. **Model/Dataset.** We study adaptation on pre-trained BERT$_{base}$ model and RoBERTa$_{large}$ hosted by HuggingFace [47]. We use the GLUE benchmark [44], which is a collection of sentence or sentence-pair natural language understanding tasks including question answering, sentiment analysis, and textual entailment. We exclude tasks that have very small datasets (e.g.,CoLA, RTE). We report the details about the hyperparameters in Appendix B.

### 4.1 Main Results – Adaptation Time Acceleration

We first compare the following schemes: (1) **Single GPU + SB:** This is the existing PyTorch implementation of Transformer fine-tuning from HuggingFace (HF), using small batch (SB) sizes (e.g., 32). (2) **Multi-GPU + SB:** This is multi-GPU PyTorch implementation using DistributedDataParallel [27],

(3) **Multi-GPU + LB + FreeLB:**, this is the work described in [58] using large minibatches (LB),
e.g., 1K, and perturbation step $K = 5$ for adaptation, and (4) **Multi-GPU + LB + ScaLA:** This is our
approach as described in Algorithm 1. Table 1 shows results on MNLI, QNLI, QQP, and SST2, which
are larger datasets and less sensitive to random seeds. $n \times g$ refers to $P_n$ nodes each with $P_g$ GPUs
for a total of $P = P_n P_g$ homogeneous workers (e.g., 32 GPUs on 2 NVIDIA DGX-2 nodes). For a
fair comparison, we reproduce BERT and RoBERTa baseline. Our reproduced baseline achieves the
same or slightly higher accuracy than the originally reported results in [9] and [30]. We now discuss
our results and observations.

Table 1: The adaptation time and accuracy results on GLUE benchmark. ScaLA achieves the same
average accuracy as the baseline while providing up to $18\times$ speedups than single GPU, and up to
$9.8\times$ speedups with the same amount of hardware.

| BERT$_{base}$ | n×g | bsz | MNLI-m | | | QNLI | | | QQP | | | SST-2 | | | Avg. |
| | | | Steps | Time | Acc. | Steps | Time | Acc. | Steps | Time | Acc/F1 | Steps | Time | Acc. | |
|---|---|---|---|---|---|---|---|---|---|---|---|---|---|---|---|
| Devlin et al. 2019 | | | | | 84.4 | | | 88.4 | | | - | | | 92.7 | - |
| Baseline (B=32) | 1x1 | 32 | 73632 | 19635 | 84.8 | 19644 | 5535 | 90.6 | 68226 | 16494 | 91/88.0 | 12630 | 2736 | 93.1 | 89.4 |
| Baseline (B=32) | 2x16 | 32 | 73632 | 8848 | 84.8 | 19644 | 2408 | 90.6 | 68226 | 11311 | 91/88.0 | 12630 | 1494 | 93.1 | 89.4 |
| FreeLb (B=1K) | 2x16 | 1K | 2301 | 5953 | 85.2 | 615 | 1944 | 90.3 | 2133 | 19030 | 91.2/88.2 | 396 | 680 | 92.8 | **89.5** |
| ScaLA (B=1K) | 2x16 | 1K | 2301 | **1323** | 85.1 | 615 | **432** | 90.0 | 2133 | **4229** | 90.9/87.7 | 396 | **151** | 93.5 | 89.4 |

| RoBERTa$_{large}$ | n×g | bsz | MNLI-m | | | QNLI | | | QQP | | | SST-2 | | | Avg. |
| | | | Steps | Time | Acc. | Steps | Time | Acc. | Steps | Time | Acc/F1 | Steps | Time | Acc. | |
|---|---|---|---|---|---|---|---|---|---|---|---|---|---|---|---|
| Liu et al. 2020 | | | | | 90.2 | | | 94.7 | | | 92.2/- | | | 96.4 | - |
| Baseline (B=32) | 1x1 | 32 | 73632 | 43090 | 90.5 | 19644 | 14188 | 94.7 | 68226 | 40945 | 92.0/89.4 | 12630 | 4940 | 96.4 | 92.5 |
| Baseline (B=32) | 2x16 | 32 | 73632 | 18114 | 90.5 | 19644 | 4842 | 94.7 | 68226 | 16614 | 92.0/89.4 | 12630 | 3072 | 96.4 | 92.5 |
| FreeLb (B=1K) | 2x16 | 1K | 2301 | 15133 | 91.2 | 615 | 5256 | 95.2 | 2133 | 10818 | 92.5/90.0 | 396 | 1804 | 96.9 | **93.3** |
| ScaLA (B=1K) | 2x16 | 1K | 2301 | **3363** | 90.9 | 615 | **1168** | 95.1 | 2133 | **2404** | 92.3/89.8 | 396 | **401** | 96.7 | 92.9 |

**Adaptation time analysis.** Compared with single-GPU training, the multi-GPU baseline leads to
only modest training speedup improvements, e.g., with $1.5 - 2.4\times$ faster training speed for both
BERT and RoBERTa, even with $32\times$ more compute resources. The speedup is limited because of the
small mini-batches (e.g., 32) used for adaptation, which do not provide a sufficient workload to fully
utilize the underlying hardware. Thus, communication overhead becomes the dominant part, and the
adaptation often struggles to obtain speedups even with more workers. In contrast, ScaLA achieves
up to $18\times$ speedups over the single-GPU baseline with 32 GPUs. When using the same number of
GPUs (e.g., 32), ScaLA is 2.7–9.8$\times$ faster. The speedups come from three aspects: (1) the improved
hardware efficiency for each worker from increased per-worker micro-batch size; (2) the reduced
all-reduce communication overhead since it takes fewer communication rounds to process the same
number of samples in one epoch; (3) the lightweight adversarial noise incurs only a small portion of
the total training overhead. Finally, ScaLA obtains the speedups while achieving the same accuracy
(88.4 vs. 88.4) average accuracy for BERT and higher accuracy (92.9 vs. 92.5) for RoBERTa as the
baselines. ScaLA is 4.5 times faster than FreeLb while achieving similar accuracy on BERT (89.4 vs.
89.5) and RoBERTa (92.9 vs. 93.5). ScaLA is faster than FreeLb because FreeLb does not consider
much about the training cost and performs multiple ascent steps to calculate adversaries across the full
training process. As a matter of fact, FreeLb is even slower to run than vanilla baseline (e.g., QNLI
on RoBERTa). In contrast, ScaLA analyzes the computational efficiency of adversarial large-batch
optimization and introduces several simple yet effective approaches to reduce the adversarial noise
cost, which leads to overall improved computational efficiency.

**Generalizability analysis.** Since there are very few works on large-batch adaptation, we create
several baselines to compare with ScaLA: (1) Multi-GPU + LB + Tuning LR: This configuration uses
large mini-batches (e.g., 1K), and applies heuristic-based scheduling rule (e.g., square root) combined
with an extensive grid search for learning rates; (2) Multi-GPU + LB + LAMB: Uses LAMB [52]
optimizer for large-batch adaptation. We make several observations from the results in Table 2. First,
compared with the baseline accuracy reported in the paper, the accuracy of Multi-GPU + LB drops
by close to 1 point (88.4 vs. 89.4, and 92.1 vs. 92.9) in average and close to 2 points for some tasks
(e.g., QQP on BERT), indicating that it is challenging to obtain on-par accuracy with large-batch
optimizations for adaptation despite with heavy hyperparameter tuning. Second, since LAMB is
designed primarily for improving the convergence of pre-training instead of the adaptation, its ability
to accelerate the adaptation has yet to be proven. In our experiments, LAMB leads to only marginal
improvements (88.6 vs. 88.4, and 92.1 vs. 92.1) than the baseline and is 0.8 points lower than the
small-batch baseline. This is because LAMM does not directly minimize the sharpness of the loss
landscape, so it can still lead to poor generalizability during adaptation. With ScaLA, we are able

to close the generalization gap from large-batch optimization (89.4 vs. 89.4, and 92.5 vs. 92.9) and achieve 0.8 points higher accuracy (89.4 vs. 88.6, 92.9 vs. 92.1) than LAMB on both BERT and RoBERTa. ScaLA improves generalizability because it introduces adversarial noise in the large-batch optimization process, which serves as a regularizer. By training the network to be robust to such perturbations, the model loss landscape is smoothed out, leading to improved generalization.

Table 2: The comparison results between ScaLA and alternative methods for large-batch adaptation on the GLUE benchmark, which show that ScaLA achieves higher accuracy than baselines after training the same number of samples and steps.

| BERT$_{base}$ | n×g | Batch size | MNLI-m | | | QNLI | | | QQP | | | SST-2 | | | Avg. |
| | | | Steps | Time | Acc. | Steps | Time | Acc. | Steps | Time | Acc./F1 | Steps | Time | Acc. | |
| Vanilla (B=1K) | 2x16 | 1K | 2301 | 1148 | 84.3 | 615 | 349 | 89.3 | 2133 | 2892 | 89.6/86.1 | 396 | 134 | 93 | 88.4 |
| LAMB (B=1K) | 2x16 | 1K | 2301 | 1180 | 84.1 | 615 | 359 | 89.6 | 2133 | 2978 | 90.5/87.0 | 396 | 139 | 92.4 | 88.6 |
| ScaLA (B=1K) | 2x16 | 1K | 2301 | 1323 | **85.1** | 615 | 432 | **90.0** | 2133 | 4229 | **90.9/87.7** | 396 | 151 | **93.5** | **89.4** |

| RoBERTa$_{large}$ | n×g | Batch size | MNLI-m | | | QNLI | | | QQP | | | SST-2 | | | Avg. |
| | | | Steps | Time | Acc. | Steps | Time | Acc. | Steps | Time | Acc./F1 | Steps | Time | Acc. | |
| Vanilla (B=1K) | 2x16 | 1K | 2301 | 2514 | 90.1 | 615 | 936 | 94.3 | 2133 | 1874 | 91.7/89.1 | 396 | 317 | 95.9 | 92.1 |
| LAMB (B=1K) | 2x16 | 1K | 2301 | 2646 | 90.5 | 615 | 973 | 94.5 | 2133 | 1998 | 91.3/88.5 | 396 | 324 | 96.2 | 92.1 |
| ScaLA (B=1K) | 2x16 | 1K | 2301 | 3363 | **90.9** | 615 | 1168 | **95.1** | 2133 | 2404 | **92.3/89.8** | 396 | 401 | **96.7** | **92.9** |

## 4.2   Experiment – Analysis Results

**Ablation analysis:** We study the importance of components in ScaLA. We set $t_s$ to 0, which denotes as *w/o Delaying PGA-1*. We replace the outer minimization to use ADAM [26], which is noted as *w/o Groupwise LR*. We set $\lambda$ to 0, which denotes as *w/o PGA-1*. The results are reported in Table 3.

Table 3: Ablation study of ScaLA using BERT$_{base}$ on GLUE tasks.

| | MNLI-m | | QNLI | | QQP | | SST-2 | | Avg. | Speedup |
| | Time | Acc. | Time | Acc. | Time | Acc./F1 | Time | Acc. | | |
| Baseline | 19635 | 84.8 | 5535 | 90.6 | 16494 | 91/88.0 | 2736 | 93.1 | 89.4 | 1 |
| ScaLA | 1323 | 85.1 | 432 | 90 | 4229 | 90.9/87.7 | 151 | 93.5 | 89.4 | 12.4 |
| w/o Delaying PGA-1 | 2503 | 85.2 | 726 | 90.2 | 6407 | 91.3/88.3 | 272 | 93.1 | 89.5 | 7.0 |
| w/o Groupwise LR | 1290 | 85.0 | 422 | 89.9 | 4212 | 90.7/87.6 | 146 | 93.0 | 89.2 | 12.7 |
| w/o PGA-1 | 1180 | 84.1 | 359 | 89.6 | 2978 | 90.5/87.0 | 139 | 92.4 | 88.6 | 14.3 |

The results in Table 3 show that the removal of either design element would result in a performance drop. For example, removing PGA-1 leads to 0.8 points accuracy drop (88.6 vs. 89.4), indicating that adversarial noise is crucial for improving the generalizability of large-batch adaptation. Moreover, if we perform PGA-1 without delayed injection, the average accuracy increases by 0.1 points (89.5 vs. 89.4), but the execution time is increased by 1.5–1.9x, indicating the importance of having lightweight adversarial noise for speeding up the adaptation. Finally, removing group-wise learning rates leads to a small 0.2 points accuracy drop (89.2 vs. 89.4), indicating that ScaLA still achieves benefits without group-wise learning rates (89.2 vs. 88.6), but they are complementary to each other.

Table 4: Alternatives to generate perturbations using random noise, ground-truth, and label probability.

| Model | MNLI-m | QNLI | QQP | SST-2 | Avg |
| Baseline | 84.3 | 89.3 | 89.6/86.1 | 93 | 88.4 |
| Gaussian noise | 84.5 | 89.4 | 90.3/87.0 | 92.6 | 88.7 |
| ScaLA (GT) | 84.1 | 89.6 | 90.7/87.6 | 93.2 | 89.0 |
| ScaLA (LP) | **85.1** | 90 | **90.9/87.7** | **93.5** | **89.4** |

**Curvature analysis.** We measure the steepness of the loss landscape again after applying ScaLA. As shown in Fig. 4(c), the largest eigenvalue of the model becomes much smaller (6.9×) with lower deviations with ScaLA and is slightly better than the small batch baseline, which is a strong indication that our approach enforces the smoothness of the model that leads to the accuracy improvement.

**Comparison with random noise.** We have performed additional experiments by adding Gaussian noise to the embeddings. Table 4 that random noise indeed can improve the accuracy for MNLI-m (84.3 vs. 84.5), QNLI (89.3 vs. 89.4), and QQP (90.3/87.0 vs. 89.6/86.1) over the baseline, but it also leads to worse results on SST-2 (93. vs. 92.6). Compared with ScaLA, random noise consistently falls

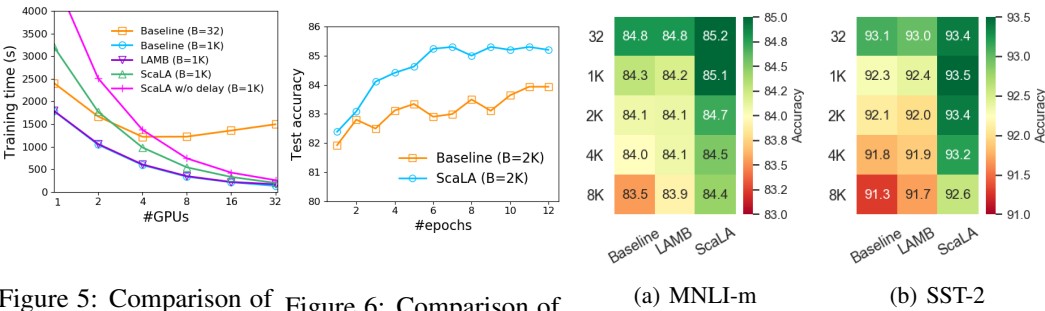

Figure 5: Comparison of scalability using different large-batch optimization methods on SST-2.

Figure 6: Comparison of test accuracy by training the baseline longer.

(a) MNLI-m      (b) SST-2

Figure 7: Comparison of accuracy under even larger batch sizes.

behind ScaLA in its ability to reduce the generalization error on all tested tasks and is on average 0.7 points lower than ScaLA (88.7 vs. 89.4). These results indicate that ScaLA's approach of explicitly enforcing the smoothness of the loss landscape can result in better improvement.

**Perturbations via ground-truth vs. label probability.** We also create one-hot labels and use those to generate perturbations instead of using label probability generated by the network. Table 4 shows that using label probability (LP) consistently leads to higher accuracy than using the ground-truth (GT), e.g., 89.4 vs. 89.0 on average. Label probability leads to better generalization, probably because it provides a better measurement of the adversarial direction, which is the direction in the input space in which the label probability of the model is most sensitive to small perturbations.

**Scalability analysis varying GPUs.** Figure 5 shows the scalability comparison on SST-2 after optimizations. While the speedup still plateaus at 4 GPUs with a small batch size (e.g., $B = 32$), the four large-batch configurations are able to scale well up to 32 GPUs and take a similar amount of time with 32 GPUs. ScaLA scales better than ScaLA without delaying PGA-1, and achieves a much faster training speed, especially in the 1-16 GPU range.

**Train longer, generalize better?** Despite improved adaptation speed, one may still wonder whether simply performing large-batch adaptation longer would also close the generalization gap. Figure 6 shows the comparison between ScaLA and the baseline on a batch size of 2K. ScaLA obtains an accuracy of 85.2 after 6 epochs of training, whereas the baseline has difficulty to reach 84 after training twice longer (e.g., 12 epochs). ScaLA achieves better accuracy because it explicitly penalizes model weights from getting stuck at sharp minima, leading to better generalizability.

**Generalizability under different batch sizes.** We also evaluate how different batch sizes affect the generalizability of adapting transformers. Figure 7 shows the results on MNLI-m and SST-2. We make two major observations: (1) The accuracy tends to drop as the batch size increases. (2) While both the baseline and LAMB suffer from significant accuracy drop by drastically increasing the batch size (e.g., from 32 to 8K), ScaLA is able to mitigate the generalization gap and consistently achieves higher accuracy than the baseline (e.g., 84.4 vs. 83.5 for MNLI, and 92.6 vs. 91.3 for SST-2 at batch size 8K) and LAMB (e.g., 84.4 vs. 83.9 for MNLI, and 92.6 vs. 91.7 for SST-2 at batch size 8K). These results indicate the benefit of ScaLA is maintained by further increasing the batch size, which could bring even greater speedups when increasing the data parallelism degree.

## 5   Conclusions and Future Directions

In this paper, we study how to accelerate the adaptation speed of pre-trained Transformer models for NLU tasks. We introduce ScaLA, an efficient large-batch adaptation method using carefully injected lightweight adversarial noises. The experiment results show that ScaLA obtains up to $9.8\times$ speedups on adapting transformer networks and outperforms state-of-the-art large-batch optimization methods in generalizability. Given the promising results of ScaLA on accelerating the adaptation speed, it opens new research opportunities on applying ScaLA to accelerate the more expensive pre-training tasks as well as emerging pre-trained transformer networks for computer vision domains tasks.

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
