# A    Additional Analysis Results

In the part, we present results that are not included in the main text due to the space limit.

## A.1    The Usefulness of Adversarial Noises at Different Epochs

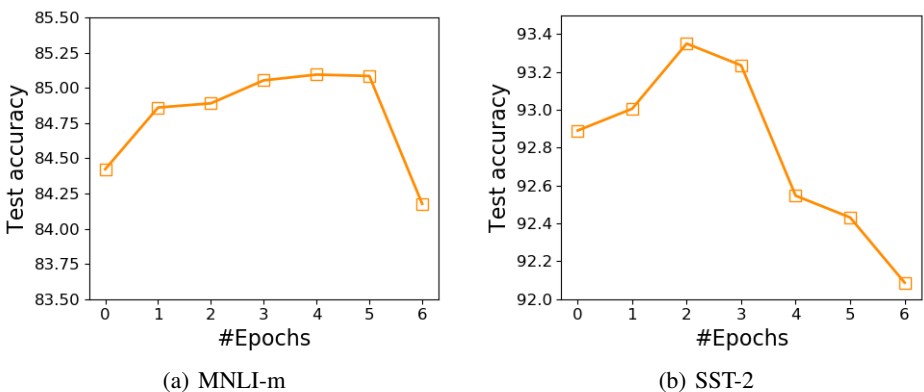

(a) MNLI-m          (b) SST-2

Figure 8: Accuracy results from delaying the injection of adversarial noises at different epochs.

In Section 3.1, we mention that no adversaries are needed at the initial epochs of adaptation. To verify, we conduct experiments to measure the final accuracy corresponding to starting from regular training and switching to PGA-1 after $t_s$ epochs, where $t_s \in [T]$. Figure 8 shows that enabling PGA-1 from the very beginning does not offer much improvement on accuracy. However, as we delay the injection of adversarial noises, the model accuracy starts to increase. By delaying the injection of adversarial noises, we observe improved test accuracy on downstream tasks. However, it also seems that the adversarial noise should not be injected too late, which may inadvertently affect the accuracy. It is possible that a more advanced method to adaptively choose the value of $t_s$ is desired. However, given that (1) the primary focus of this work is to demonstrate that it is possible and effective to accelerate the adaptation of transformer networks via large-batch adaptation and adversarial noises and (2) the search space of is quite small for most downstream tasks, we leave this as an interesting research question for future exploration.

# B    Hyperparameters

For all configurations, we fine-tune against the GLUE datasets and set the maximum number of epochs to 6. We use a linear learning rate decay schedule with a warm-up ratio of 0.1. For ScaLA, we set $\lambda = 1$, perturbation clipping radius $\omega = 10^{-5}$, step size $\rho = 10^{-4}$, and $t_s=\{3,5\}$. These values worked well enough that we did not feel the need to explore more. For fairness, we perform a grid search of learning rates in the range of $\{$1e-5, 3e-5, 5e-5, 7e-5, 9e-5, 1e-4, 3e-4$\}$ for small batch sizes and $\{$5.6e-5, 8e-5, 1e-4, 1.7e-4, 2.4e-4, 2.8e-4, 4e-4, 5.6e-4, 1e-3$\}$ for large batch sizes. We keep the remaining hyperparameteres unchanged.

# C    Hyperparameter Tuning Cost for Large-Batch Adaptation with ScaLA

In this part, we investigate how large-batch adaptation affects the generalizability of transformer networks on downstream tasks. As there are various heuristics for setting the learning rates [42, 14, 41, 52], and because few work studies the learning rate scaling effects on adapting pre-trained Transformer networks, we perform a grid search on learning rates $\{$1e-4, 3e-4,5e-4, 7e-4, 9e-4, 1e-3, 3e-3$\}$ and batch sizes $\{$1K, 2K, 4K, 8K$\}$ while keeping the other hyperparameters the same to investigate how ScaLA affects the hyperparameter tuning effort.

Table 5 shows the results of using the square root scaling rule to decide the learning rates for large batch sizes vs. accuracy results with tuned learning rate results, without and with ScaLA. The first row represents the best accuracy found through fine-tuning with a small batch size 32. The next

two rows correspond to fine-tuning with batch size 1024 using tuned learning rates vs. using the scaling rule. The last two rows represent fine-tuning using ScaLA with batch size 1024, also using tuned learning rates vs. the scaling rule. Even with square-root scaling, the large-batch baseline still cannot reach the small-batch accuracy (88.7 vs. 89.4). Moreover, although tuning the learning rates lead to better results on some datasets such as MNLI-m (84.9 vs. 85.1) and SST-2 (92.9 vs. 93.5), the square-root scaling rule leads to better results on other tasks such as QNLI (90.8 vs. 90) and QQP (91.4/88.4 vs. 90.9/87.7). So the best learning rates on fine-tuning tasks are not exactly sqrt. However, given that ScaLA with square-root learning rate scaling achieves on average better results than the grid search of learning rates (89.4 vs. 89.7), we suggest to use sqrt scaling for learning rates to simplify the hyperparameter tuning effort for ScaLA.

Table 5: Evaluation results on hyperparameter tuning vs. using square-root learning rate scaling.

| | MNLI-m | QNLI | QQP | SST-2 | Avg |
|---|---|---|---|---|---|
| Bsz=32 (tuned, baseline) | 84.8 | 90.6 | 91/88 | 93.1 | 89.4 |
| Bsz=1024 (tuned, baseline) | 84.3 | 89.3 | 89.6/86.1 | 93 | 88.5 |
| Bsz=1024 (scaling rule, baseline) | 83.9 | 89.2 | 90.6/87.4 | 92.5 | 88.7 |
| Bsz=1024 (tuned, ScaLA) | 85.1 | 90 | 90.9/87.7 | 93.5 | 89.4 |
| Bsz=1024 (scaling rule, ScaLA) | 84.9 | 90.8 | 91.4/88.4 | 92.9 | 89.7 |

## D  Convergence Analysis

In this section, we provide the formal statements and detailed proofs for the convergence rate. The convergence analysis builds on techniques and results in [7, 53]. We consider the general problem of a two-player sequential game represented as nonconvex-nonconcave minimax optimization that is stochastic with respect to the outer (first) player playing $x \in \mathbb{X}$ while sampling $\xi$ from $Q$ and deterministic with respect to the inner (second) player playing $y \in \mathbb{Y}$, i.e.,

$$\min_x \max_y \mathbb{E}_{\xi \sim Q}[f(x, y, \xi)] := \min_x \mathbb{E}_{\xi \sim Q}[g(x, \xi)] \tag{3}$$

Since finding the Stackelberg equilibrium, i.e., the global solution to the saddle point problem, is NP-hard, we consider the optimality notion of a *local minimax* point [23]. Since maximizing over $y$ may result in a non-smooth function even when $f$ is smooth, the norm of the gradient is not particularly a suitable metric to track the convergence progress of an iterative minimax optimization procedure. Hence, we use the gradient of the *Moreau envelope* [8] as the appropriate potential function. Let $\mu \in \mathbb{R}^h_+$. The $\mu$-Moreau envelope for a function $g : \mathbb{X} \to \mathbb{R}$ is defined as $g_\mu(x) := \min_z g(z) + \sum_{i=1}^h \frac{1}{2\mu^i} \|x^i - z^i\|^2$. Another reason for the choice of this potential function is due to the special property [38] of the Moreau envelope that if its gradient $\nabla_x[g_\mu(x)]$ almost vanishes at $x$, such $x$ is close to a stationary point of the original function $g$.

**Assumptions:**  We assume that $\mathbb{X} = \bigsqcup_{i=1}^h \mathbb{X}^i$ is partitioned into $h$ disjoint groups , i.e., in terms of training a neural network, we can think of the network having the parameters partitioned into $h$ (hidden) layers. The measure $Q$ characterizes the training data. Let $\widehat{\nabla}_x f(x, y)$ denote the noisy estimate of the true gradient $\nabla_x f(x, y)$. We assume that the noisy gradients are unbiased, i.e., $\mathbb{E}[\widehat{\nabla}_x f(x, y)] = \nabla_x f(x, y)$. For each group $i \in [h]$, we make the standard (groupwise) boundedness assumption [11] on the variance of the stochastic gradients, i.e., $\mathbb{E}\|\widehat{\nabla}_x^i f(x, y) - \nabla_x^i f(x, y)\|^2 \le \sigma_i^2$, $\forall i \in [h]$. We assume that $f(x, y)$ has Lipschitz continuous gradients. Specifically, let $f(x, y)$ be $\alpha$-smooth in $x$ where $\alpha := (\alpha_1, \ldots, \alpha_h)$ denotes the $h$-dimensional vector of (groupwise) Lipschitz parameters, i.e., $\|\nabla_x^i f(x_a, y) - \nabla_x^i f(x_b, y)\| \le \alpha_i \|x_a^i - x_b^i\|$, $\forall i \in [h]$ and $x_a, x_b \in \mathbb{X}, y \in \mathbb{Y}$. Let $\kappa_\alpha := \frac{\max_i \alpha_i}{\min_i \alpha_i}$.

Super-scripts are used to index into a vector ($i$ denotes the group index and $j$ denotes an element in group $i$). For any $c \in \mathbb{R}$, the function $\nu : \mathbb{R} \to [\mathcal{L}, \mathcal{U}]$ clips its values, i.e., $\nu(c) := \max(\mathcal{L}, \min(c, \mathcal{U}))$ where $\mathcal{L} < \mathcal{U}$. Let $\|.\|, \|.\|_1$ and $\|.\|_\infty$ denote the $\ell_2, \ell_1$, and $\ell_\infty$ norms. We assume that the true gradients are bounded, i.e., $\|\nabla_x f(x, y)\|_\infty \le \mathcal{G}$.

First, we begin with relevant supporting lemmas. The following lemma characterizes the convexity of an additive modification of $g$.

**Lemma D.1** ([28, 23, 36]). *Let $g(x) := \max_y f(x, y)$ with $f$ being $\alpha$-smooth in $x$ where $\alpha \in \mathbb{R}^h_+$ is the vector of groupwise Lipschitz parameters. Then, $g(x) + \sum_{i=1}^h \frac{\alpha_i}{2} \|x^i\|^2$ is convex in $x$.*

The following property of the Moreau envelope relates it to the original function.

**Lemma D.2** ([38]). *Let $g$ be defined as in Lemma D.1. Let $\widehat{x} = \arg\min_{\widetilde{x}} g(\widetilde{x}) + \sum_{i=1}^h \frac{1}{2\mu^i} \|\widetilde{x}^i - x^i\|^2$. Then, $\|g_\mu(x)\| \le \epsilon$ implies $\|\widehat{x} - x\| \le \|\mu\|_\infty \epsilon$ and $\min_h \|h\| \le \epsilon$ with $h \in \partial g$ where $\partial g$ denotes the subdifferential of $g$.*

We now present the formal version of Theorem 3.1 in Theorem D.3. Note that Lemma D.2 facilitates giving the convergence guarantees in terms of the gradient of the Moreau envelope. Recall that $t \in [T]$ denotes the epochs corresponding to the outer maximization. Without loss of generality, we

697  set the delay parameter for injection of the adversarial perturbation in Algorithm 1 as $t_s = 0$. Here,
698  we assume that the PGA provides an $\epsilon$-approximate maximizer.

**Theorem D.3** (Groupwise outer minimization with an $\epsilon$-approximate inner maximization oracle).
*Let us define relevant constants as $\mathcal{D} := \left(g_{1/2\alpha}(x_0) - \mathbb{E}(\min_x g(x))\right)$ being the optimality gap due
to initialization, $\kappa_\alpha := \frac{\max_i \alpha_i}{\min_i \alpha_i}$ being the condition number, $\|\nabla_x f(x,y)\|_\infty \leq \mathcal{G}$ being gradient
bound, $\mathcal{Z} := \max_{i,j,t} \frac{(\hat{x}_t^{i,j} - x_t^{i,j})}{(\nabla_t^{i,j})} \sigma_i$ being the variance term, $\mathcal{L}, \mathcal{U}$ being clipping constants such that
$\mathcal{L} \leq \mathcal{U}$. For the outer optimization, setting the learning rate as $\eta = \frac{1}{\mathcal{U}\sqrt{T}}$ and scaling batch size as
$b = \frac{16 T \mathcal{L}^2 \mathcal{Z}^2}{\mathcal{U}^2}$, we have*

$$\mathbb{E}\left[\|\nabla g_{1/2\alpha}(\overline{x})\|^2\right] \leq 4\epsilon\|\alpha\|_\infty + \frac{2\kappa_\alpha \mathcal{D}\mathcal{G}}{\sqrt{T}} \tag{4}$$

705  *where $\overline{x}$ is the estimator obtained from running $T$ steps of Algorithm 1 and picking $x_t$ uniformly at
706  random for $t \in [T]$.*

707  *Proof.* In this proof, for brevity, we define the vector $\nabla_t := \nabla_x f(x,y)$, i.e., the gradient of the
708  objective with respect to $x$, evaluated at the outer step $t$. Since evaluating gradients using mini-batches
709  produces noisy gradients, we use $\widehat{\nabla}$ to denote the noisy version of a true gradient $\nabla$, i.e., $\widehat{\nabla} = \nabla + \Delta$
710  for a noise vector $\Delta$. For any outer step $t$, we have $f(x_t, \widehat{y}) \geq g(x_t) - \epsilon$ where $\widehat{y}$ is an $\epsilon$-approximate
711  maximizer. For any $\widetilde{x} \in \mathbb{X}$, using the smoothness property (Lipschitz gradient) of $f$, we have

$$g(\widetilde{x}) \geq f(\widetilde{x}, y_t)$$
$$\geq f(x_t, y_t) + \sum_{i=1}^{h} \langle \nabla_t^i, \widetilde{x}^i - x_t^i \rangle - \sum_{i=1}^{h} \frac{\alpha_i}{2} \|\widetilde{x}^i - x_t^i\|^2$$
$$\geq g(x_t) - \epsilon + \sum_{i=1}^{h} \langle \nabla_t^i, \widetilde{x}^i - x_t^i \rangle - \sum_{i=1}^{h} \frac{\alpha_i}{2} \|\widetilde{x}^i - x_t^i\|^2 \tag{5}$$

712  Let $\phi_\mu(x, z) := g(z) + \sum_{i=1}^{h} \frac{1}{2\mu^i} \|x^i - z^i\|^2$. Recall that the $\mu$-Moreau envelope for $g$ is defined as
713  $g_\mu(x) := \min_z \phi_\mu(x, z)$ and its gradient is the groupwise proximal operator given by $\nabla_x[g_\mu(x)] =$
714  $\left[\frac{1}{\mu^1}\left(x^1 - \arg\min_{z^1} \phi_\mu(x, z)\right), \ldots, \frac{1}{\mu^h}\left(x^h - \arg\min_{z^h} \phi_\mu(x, z)\right)\right]$.

715  Now, let $\widehat{x}_t = \arg\min_x \phi_{1/2\alpha}(x_t, x) = \arg\min_x \left(g(x) + \sum_{i=1}^{h} \alpha_i \|x_t^i - x^i\|^2\right)$. Then, plugging
716  in the update rule for $x$ at step $t+1$ in terms of quantities at step $t$, using the shorthand $\nu_t^i := \nu(\|x_t^i\|)$

 and conditioning on the filtration up to time $t$, we have

$$g_{1/2\alpha}(x_{t+1}) \le g(\widehat{x}_t) + \sum_{i=1}^{h} \alpha_i \|x_{t+1}^i - \widehat{x}_t^i\|^2$$

$$\le g(\widehat{x}_t) + \sum_{i=1}^{h} \alpha_i \left\| x_t^i - \eta_t \nu_t^i \frac{\widehat{\nabla}_t^i}{\|\widehat{\nabla}_t^i\|} - \widehat{x}_t^i \right\|^2$$

$$\le g(\widehat{x}_t) + \sum_{i=1}^{h} \alpha_i \left\| x_t^i - \widehat{x}_t^i \right\|^2 + \sum_{i=1}^{h} 2\alpha_i \eta_t \left\langle \nu_t^i \frac{\widehat{\nabla}_t^i}{\|\widehat{\nabla}_t^i\|}, \widehat{x}_t^i - x_t^i \right\rangle + \sum_{i=1}^{h} \alpha_i \eta_t^2 (\nu_t^i)^2$$

$$\le g_{1/2\alpha}(x_t) + \sum_{i=1}^{h} 2\alpha_i \eta_t \left\langle \nu_t^i \frac{\widehat{\nabla}_t^i}{\|\widehat{\nabla}_t^i\|}, \widehat{x}_t^i - x_t^i \right\rangle + \sum_{i=1}^{h} \alpha_i \eta_t^2 (\nu_t^i)^2$$

$$\le g_{1/2\alpha}(x_t) + 2\eta_t \sum_{i=1}^{h} \alpha_i \nu_t^i \sum_{j=1}^{d_i} \left( \frac{\widehat{\nabla}_t^{i,j}}{\|\widehat{\nabla}_t^i\|} - \frac{\nabla_t^{i,j}}{\|\nabla_t^i\|} + \frac{\nabla_t^{i,j}}{\|\nabla_t^i\|} \right) \times (\widehat{x}_t^{i,j} - x_t^{i,j}) + \sum_{i=1}^{h} \alpha_i \eta_t^2 (\nu_t^i)^2$$

$$\le g_{1/2\alpha}(x_t) + 2\eta_t \sum_{i=1}^{h} \alpha_i \nu_t^i \sum_{j=1}^{d_i} \left( \frac{\nabla_t^{i,j}}{\|\nabla_t^i\|} \right) \times (\widehat{x}_t^{i,j} - x_t^{i,j})$$

$$+ 2\eta_t \sum_{i=1}^{h} \alpha_i \nu_t^i \sum_{j=1}^{d_i} \left( \frac{\widehat{\nabla}_t^{i,j}}{\|\widehat{\nabla}_t^i\|} - \frac{\nabla_t^{i,j}}{\|\nabla_t^i\|} \right) \times (\widehat{x}_t^{i,j} - x_t^{i,j}) + \sum_{i=1}^{h} \alpha_i \eta_t^2 (\nu_t^i)^2$$

$$\le g_{1/2\alpha}(x_t) + 2\eta_t \sum_{i=1}^{h} \frac{\alpha_i \nu_t^i}{\|\nabla_t^i\|} \left\langle \nabla_t^i, \widehat{x}_t^i - x_t^i \right\rangle$$

$$+ 2\eta_t \sum_{i=1}^{h} \alpha_i \nu_t^i \sum_{j=1}^{d_i} \left( \frac{\nabla_t^{i,j} + \Delta_t^{i,j}}{\|\nabla_t^i + \Delta_t^i\|} - \frac{\nabla_t^{i,j}}{\|\nabla_t^i\|} \right) \times (\widehat{x}_t^{i,j} - x_t^{i,j}) + \sum_{i=1}^{h} \alpha_i \eta_t^2 (\nu_t^i)^2$$

$$\leq g_{1/2\alpha}(x_t) + 2\eta_t \mathcal{U} \sum_{i=1}^{h} \frac{\alpha_i}{\|\nabla_t^i\|} \left\langle \nabla_t^i, \widehat{x}_t^i - x_t^i \right\rangle$$

$$+ 2\eta_t \sum_{i=1}^{h} \alpha_i \nu_t^i \sum_{j=1}^{d_i} \left( \frac{\|\nabla_t^i\|(\nabla_t^{i,j})(\nabla_t^{i,j} + \Delta_t^{i,j}) - \|\nabla_t^i + \Delta_t^i\|(\nabla_t^{i,j})^2}{\|\nabla_t^i + \Delta_t^i\|\|\nabla_t^i\|} \right) \times \frac{(\widehat{x}_t^{i,j} - x_t^{i,j})}{(\nabla_t^{i,j})}$$

$$+ \sum_{i=1}^{h} \alpha_i \eta_t^2 (\nu_t^i)^2$$

$$\overset{E_1}{\leq} g_{1/2\alpha}(x_t) + 2\eta_t \mathcal{U} \max_i \frac{\alpha_i}{\|\nabla_t^i\|} \left( g(\widehat{x}_t) - g(x_t) + \epsilon + \sum_{i=1}^{h} \frac{\alpha_i}{2} \|\widehat{x}^i - x_t^i\|^2 \right)$$

$$+ 2\eta_t \sum_{i=1}^{h} \alpha_i \nu_t^i \max_j \frac{(\widehat{x}_t^{i,j} - x_t^{i,j})}{(\nabla_t^{i,j})} \left( \frac{\langle \nabla_t^i, \nabla_t^i + \Delta_t^i \rangle - \|\nabla_t^i + \Delta_t^i\|\|\nabla_t^i\|}{\|\nabla_t^i + \Delta_t^i\|} \right) + \sum_{i=1}^{h} \alpha_i \eta_t^2 (\nu_t^i)^2$$

$$\leq g_{1/2\alpha}(x_t) + 2\eta_t \mathcal{U} \max_i \frac{\alpha_i}{\|\nabla_t^i\|} \left( g(\widehat{x}_t) - g(x_t) + \epsilon + \sum_{i=1}^{h} \frac{\alpha_i}{2} \|\widehat{x}^i - x_t^i\|^2 \right)$$

$$- 2\eta_t \sum_{i=1}^{h} \alpha_i \nu_t^i \max_j \frac{(\widehat{x}_t^{i,j} - x_t^{i,j})}{(\nabla_t^{i,j})} \left( \frac{\|\nabla_t^i + \Delta_t^i\|\|\nabla_t^i\| - \|\nabla_t^i + \Delta_t^i\|^2 + \langle \Delta_t^i, \nabla_t^i + \Delta_t^i \rangle}{\|\nabla_t^i + \Delta_t^i\|} \right)$$

$$+ \sum_{i=1}^{h} \alpha_i \eta_t^2 (\nu_t^i)^2 \tag{6}$$

$$\leq g_{1/2\alpha}(x_t) + 2\eta_t \mathcal{U} \max_i \frac{\alpha_i}{\|\nabla_t^i\|} \left( g(\widehat{x}_t) - g(x_t) + \epsilon + \sum_{i=1}^{h} \frac{\alpha_i}{2} \|\widehat{x}^i - x_t^i\|^2 \right)$$

$$- 2\eta_t \sum_{i=1}^{h} \alpha_i \nu_t^i \max_j \frac{(\widehat{x}_t^{i,j} - x_t^{i,j})}{(\nabla_t^{i,j})} \left( \|\nabla_t^i\| - \|\nabla_t^i + \Delta_t^i\| - \frac{|\langle \Delta_t^i, \nabla_t^i + \Delta_t^i \rangle|}{\|\nabla_t^i + \Delta_t^i\|} \right) + \sum_{i=1}^{h} \alpha_i \eta_t^2 (\nu_t^i)^2$$

$$\overset{E_2}{\leq} g_{1/2\alpha}(x_t) + 2\eta_t \mathcal{U} \max_i \frac{\alpha_i}{\|\nabla_t^i\|} \left( g(\widehat{x}_t) - g(x_t) + \epsilon + \sum_{i=1}^{h} \frac{\alpha_i}{2} \|\widehat{x}^i - x_t^i\|^2 \right)$$

$$- 2\eta_t \sum_{i=1}^{h} \alpha_i \nu_t^i \max_j \frac{(\widehat{x}_t^{i,j} - x_t^{i,j})}{(\nabla_t^{i,j})} \left( \|\nabla_t^i\| - \|\nabla_t^i + \Delta_t^i\| - \|\Delta_t^i\| \right) + \sum_{i=1}^{h} \alpha_i \eta_t^2 (\nu_t^i)^2$$

$$\overset{E_3}{\leq} g_{1/2\alpha}(x_t) + 2\eta_t \mathcal{U} \max_i \frac{\alpha_i}{\|\nabla_t^i\|} \left( g(\widehat{x}_t) - g(x_t) + \epsilon + \sum_{i=1}^{h} \frac{\alpha_i}{2} \|\widehat{x}^i - x_t^i\|^2 \right)$$

$$- 4\eta_t \sum_{i=1}^{h} \alpha_i \nu_t^i \max_j \frac{(\widehat{x}_t^{i,j} - x_t^{i,j})}{(\nabla_t^{i,j})} \|\Delta_t^i\| + \sum_{i=1}^{h} \alpha_i \eta_t^2 (\nu_t^i)^2$$

$$g_{1/2\alpha}(x_T) \overset{E_4}{\leq} g_{1/2\alpha}(x_0) + 2\mathcal{U} \sum_{t=0}^{T-1} \eta_t \max_i \frac{\alpha_i}{\|\nabla_t^i\|} \left( g(\widehat{x}_t) - g(x_t) + \epsilon + \sum_{i=1}^{h} \frac{\alpha_i}{2} \|\widehat{x}^i - x_t^i\|^2 \right)$$

$$- 4 \sum_{t=0}^{T-1} \eta_t \sum_{i=1}^{h} \alpha_i \nu_t^i \max_j \frac{(\widehat{x}_t^{i,j} - x_t^{i,j})}{(\nabla_t^{i,j})} \|\Delta_t^i\| + \sum_{t=0}^{T-1} \sum_{i=1}^{h} \alpha_i \eta_t^2 (\nu_t^i)^2$$

where we have used Hölder's inequality along with bound (5) in $E_1$, Cauchy-Schwarz inequality in $E_2$, triangle inequality in $E_3$, telescoping sum in $E_4$. Rearranging and using $\eta_t = \eta$ in $E_5$ along with

Hölder's inequality,

$$\frac{1}{2\eta\mathcal{U}}\left(g_{1/2\alpha}(x_T)-g_{1/2\alpha}(x_0)\right)\le\sum_{t=0}^{T-1}\max_i\frac{\alpha_i}{\|\nabla_t^i\|}\left(g(\widehat{x}_t)-g(x_t)+\epsilon+\sum_{i=1}^{h}\frac{\alpha_i}{2}\|\widehat{x}^i-x_t^i\|^2\right)$$
$$-\frac{2}{\mathcal{U}}\sum_{t=0}^{T-1}\sum_{i=1}^{h}\alpha_i\nu_t^i\max_j\frac{(\widehat{x}_t^{i,j}-x_t^{i,j})}{(\nabla_t^{i,j})}\|\Delta_t^i\|+\frac{\eta}{2\mathcal{U}}\sum_{t=0}^{T-1}\sum_{i=1}^{h}\alpha_i(\nu_t^i)^2$$

$$\frac{1}{2\eta\mathcal{U}}\left(g_{1/2\alpha}(x_T)-g_{1/2\alpha}(x_0)\right)\overset{E_5}{\le}\max_{i,t}\frac{\alpha_i}{\|\nabla_t^i\|}\sum_{t=0}^{T-1}\left(g(\widehat{x}_t)-g(x_t)+\epsilon+\sum_{i=1}^{h}\frac{\alpha_i}{2}\|\widehat{x}^i-x_t^i\|^2\right)$$
$$-\frac{2}{\mathcal{U}}\sum_{t=0}^{T-1}\sum_{i=1}^{h}\alpha_i\nu_t^i\max_j\frac{(\widehat{x}_t^{i,j}-x_t^{i,j})}{(\nabla_t^{i,j})}\|\Delta_t^i\|+\frac{\eta}{2\mathcal{U}}\sum_{t=0}^{T-1}\sum_{i=1}^{h}\alpha_i(\nu_t^i)^2$$

Dividing by $T$ and rearranging,

$$\frac{1}{T}\sum_{t=0}^{T-1}\left(g(x_t)-g(\widehat{x}_t)-\sum_{i=1}^{h}\frac{\alpha_i}{2}\|\widehat{x}^i-x_t^i\|^2\right)\le\epsilon-\frac{1}{2\eta\mathcal{U}\zeta T}\left(g_{1/2\alpha}(x_T)-g_{1/2\alpha}(x_0)\right)$$
$$-\frac{2}{\mathcal{U}\zeta T}\sum_{t=0}^{T-1}\sum_{i=1}^{h}\alpha_i\nu_t^i\max_j\frac{(\widehat{x}_t^{i,j}-x_t^{i,j})}{(\nabla_t^{i,j})}\|\Delta_t^i\|$$
$$+\frac{\eta}{2\mathcal{U}\zeta T}\sum_{i=1}^{h}\alpha_i\sum_{t=0}^{T-1}(\nu_t^i)^2$$

where we define $\zeta:=\max_{i,t}\frac{\alpha_i}{\|\nabla_t^i\|}$. Defining $\mathcal{D}:=\left(g_{1/2\alpha}(x_0)-\mathbb{E}(\min_x g(x))\right)$ and taking expectation with respect to $\xi$ on both sides, we have

$$\frac{1}{T}\sum_{t=0}^{T-1}\mathbb{E}\left(g(x_t)-g(\widehat{x}_t)-\sum_{i=1}^{h}\frac{\alpha_i}{2}\|\widehat{x}^i-x_t^i\|^2\right)\le\epsilon+\frac{\mathcal{D}}{2\eta\mathcal{U}\zeta T}$$
$$-\frac{2\mathcal{L}}{\mathcal{U}\zeta T}\sum_{t=0}^{T-1}\sum_{i=1}^{h}\alpha_i\max_j\frac{(\widehat{x}_t^{i,j}-x_t^{i,j})}{(\nabla_t^{i,j})}\mathbb{E}\|\Delta_t^i\|+\frac{\eta\mathcal{U}\|\alpha\|_1}{2\zeta}$$
$$\overset{E_6}{\le}\epsilon+\frac{\mathcal{D}}{2\eta\mathcal{U}\zeta T}$$
$$-\frac{2\mathcal{L}}{\mathcal{U}\zeta T}\sum_{t=0}^{T-1}\sum_{i=1}^{h}\alpha_i\max_j\frac{(\widehat{x}_t^{i,j}-x_t^{i,j})}{(\nabla_t^{i,j})}\frac{\sigma_i}{\sqrt{b}}+\frac{\eta\mathcal{U}\|\alpha\|_1}{2\zeta}$$
$$\overset{E_7}{\le}\epsilon+\frac{\mathcal{D}}{2\eta\mathcal{U}\zeta T}$$
$$-\frac{2\mathcal{L}\|\alpha\|_1}{\mathcal{U}\zeta\sqrt{b}}\max_{i,j,t}\frac{(\widehat{x}_t^{i,j}-x_t^{i,j})}{(\nabla_t^{i,j})}\sigma_i+\frac{\eta\mathcal{U}\|\alpha\|_1}{2\zeta}$$
$$\overset{E_8}{=}\epsilon+\frac{\mathcal{D}}{2\eta\mathcal{U}\zeta T}-\frac{2\mathcal{L}\|\alpha\|_1\mathcal{Z}}{\mathcal{U}\zeta\sqrt{b}}+\frac{\eta\mathcal{U}\|\alpha\|_1}{2\zeta}\qquad(7)$$

where we have used the assumption on the variance of stochastic gradients in $E_6$, Hölder's inequality in $E_7$ and we define $\mathcal{Z}:=\max_{i,j,t}\frac{(\widehat{x}_t^{i,j}-x_t^{i,j})}{(\nabla_t^{i,j})}\sigma_i$ in $E_8$; $b$ denotes batch size. Now, we lower bound

the left hand side using the convexity of the additive modification of $g$.

$$g(x_t) - g(\widehat{x}_t) - \sum_{i=1}^{h} \frac{\alpha_i}{2} \|\widehat{x}^i - x_t^i\|^2$$

$$\geq g(x_t) + 0 - g(\widehat{x}_t) - \sum_{i=1}^{h} \alpha_i \|\widehat{x}^i - x_t^i\|^2 + \sum_{i=1}^{h} \frac{\alpha_i}{2} \|\widehat{x}^i - x_t^i\|^2$$

$$\geq \left( \left( g(x_t) + \sum_{i=1}^{h} \alpha_i \|x_t^i - x_t^i\|^2 \right) - \min_x \left( g(x_t) + \sum_{i=1}^{h} \alpha_i \|x^i - x_t^i\|^2 \right) \right) + \sum_{i=1}^{h} \frac{\alpha_i}{2} \|\widehat{x}^i - x_t^i\|^2$$

$$\geq \sum_{i=1}^{h} \frac{\alpha_i}{2} \|\widehat{x}^i - x_t^i\|^2 + \sum_{i=1}^{h} \frac{\alpha_i}{2} \|\widehat{x}^i - x_t^i\|^2 = \sum_{i=1}^{h} \frac{4\alpha_i^2}{4\alpha_i} \|\widehat{x}^i - x_t^i\|^2$$

$$\overset{E_9}{\geq} \frac{1}{4 \max_i \alpha_i} \|\nabla g_{1/2\alpha}(x_t)\|^2 \tag{8}$$

where we have used the expression for the gradient of the Moreau envelope in $E_9$. Combining the inequalities from Equation (8) and Equation (7), we have

$$\frac{1}{T} \sum_{t=0}^{T-1} \mathbb{E} \left( \frac{1}{4 \max_i \alpha_i} \|\nabla g_{1/2\alpha}(x_t)\|^2 \right) \leq \epsilon + \frac{\mathcal{D}}{2\eta \mathcal{U} \zeta T} + \left( \frac{\eta \mathcal{U}}{2\zeta} - \frac{2\mathcal{L}\mathcal{Z}}{\mathcal{U}\zeta\sqrt{b}} \right) \|\alpha\|_1$$

Setting the learning rate as $\eta = \frac{1}{\mathcal{U}\sqrt{T}}$ and batch size as $b = \frac{16T\mathcal{L}^2\mathcal{Z}^2}{\mathcal{U}^2}$,

$$\frac{1}{T} \sum_{t=0}^{T-1} \mathbb{E} \left[ \|\nabla g_{1/2\alpha}(x_t)\|^2 \right] \leq 4\epsilon \max_i \alpha_i + \frac{2\mathcal{D} \max_i \alpha_i}{\zeta\sqrt{T}}$$

Now, to simplify $\zeta$, using the inequality that $\max_k(a_k \cdot b_k) \geq \min_{k_a} a_{k_a} \cdot \min_{k_b} b_{k_b}$ for two finite sequences $\{a, b\}$ with positive values, along with the bounded gradients assumption, we have

$$\frac{1}{T} \sum_{t=0}^{T-1} \mathbb{E} \left[ \|\nabla g_{1/2\alpha}(x_t)\|^2 \right] \leq 4\epsilon \max_i \alpha_i + \frac{2\mathcal{D}\mathcal{G} \max_i \alpha_i}{\sqrt{T} \min_i \alpha_i} = 4\epsilon\|\alpha\|_\infty + \frac{2\kappa_\alpha \mathcal{D}\mathcal{G}}{\sqrt{T}}$$

where $\kappa_\alpha := \frac{\max_i \alpha_i}{\min_i \alpha_i}$. $\qquad\square$

In analyzing inexact version, as in Theorem D.3, we assumed the availability of an adversarial oracle. Next, we open up the adversarial oracle to characterize the oracle-free complexity. In order to do this, we will assume additional properties about the function $f$ as well as our deterministic perturbation space, $\mathbb{Y}_t \subseteq \mathbb{Y}, \forall t \in [T]$. Note that, for any given $t$, $y_\tau \in \mathbb{Y}_t, \forall \tau \in \mathcal{T}$. We recall the following guarantee for generalized non-convex projected gradient ascent.

**Lemma D.4** ([21]). *For every $t$, Let $f(x_t, \cdot)$ satisfy restricted strong convexity with parameter $\mathcal{C}$ and restricted strong smoothness with parameter $\mathcal{S}$ over a non-convex constraint set with $\mathcal{S}/\mathcal{C} < 2$, ie, $\frac{\mathcal{C}}{2}\|z - y\|^2 \leq f(x_t, y) - f(x_t, z) - \langle \nabla_z f(x_t, z), y - z \rangle \leq \frac{\mathcal{S}}{2}\|z - y\|^2$ for $y, z \in \mathbb{Y}_t$. For any given $t$, let the PGA-$\mathcal{T}$ algorithm $y_\tau \leftarrow \Pi_\epsilon[y_{\tau-1} + \rho \nabla_y f(x_t, y)]$ be executed with step size $\rho = 1/\mathcal{S}$. Then after at most $\mathcal{T} = O\left(\frac{\mathcal{C}}{2\mathcal{C}-\mathcal{S}} \log \frac{1}{\epsilon}\right)$ steps, $f(x_t, y_\mathcal{T}) \geq \max_y f(x_t, y) - \epsilon$.*

Using Theorem D.3 and Lemma D.4 (together with the additional restricted strong convexity/smoothness assumptions), we have the following theorem on the full oracle-free rates for Algorithm 1.

**Theorem D.5** (Groupwise outer minimization with inner maximization using projected gradient ascent). *Setting the inner iteration count as $\mathcal{T} = O\left(\frac{\mathcal{C}}{2\mathcal{C}-\mathcal{S}} \log \frac{8\|\alpha\|_\infty}{\epsilon}\right)$ and the outer iteration count as $T = \frac{16\kappa_\alpha \mathcal{D}^2 \mathcal{G}^2}{\epsilon^2}$, for a combined total of $O(\frac{1}{\epsilon^2} \log \frac{1}{\epsilon})$ adaptive adversarial iterations, Algorithm 1 achieves $\mathbb{E}\left[\|\nabla g_{1/2\alpha}(\overline{x})\|^2\right] \leq \epsilon$.*