# OpenReview forum: "On the Computational Efficiency of Adapting Transformer Models via Adversarial Noise"
_NeurIPS.cc/2022/Conference — NeurIPS 2022 Submitted_

### Official Review · Reviewer_oBzn · 2022-07-06

**Rating:** 6
**Confidence:** 3
**Soundness:** 3 good
**Presentation:** 3 good
**Contribution:** 3 good

**Summary:**

This paper aims to improve the efficiency of finetuning pretrained large transformers through adversarial training. It resorts to large mini-batch training and proposes several methods to reduce the computational overhead and address the optimization challenges of large mini-batch training. The paper provides an extensive empirical analysis confirming similar challenges raised by recent works. The proposed ScaLA algorithm combines several optimization tricks such as delayed and one-shot perturbations. On the BLUE benchmark, ScaLA substantially reduces the overhead of finetuning BERT and RoBERTA without hurting accuracy. Convergence analysis of the algorithm is provided. The ablation study is sound.

The paper reads like a collection of techniques mostly proposed by previous works. For example, as the paper points out, several key ideas are already established in computer vision applications, and now are adopted to textual data by this work. To be clear, I have no doubt that such efforts are of great use in real-world applications of large models. Just that I am not sure this paper is a right fit for the NeurIPS community.

====

After author response: with the additional results and analysis, as well as the experiments planned out, the paper is definitely in a better shape. I have increased my rating to 6.


**Questions:**

- The notation around Eqs. 1 and 2 is confusing to me. What are $x$, $y$, $Q$, etc.? Shouldn’t $r$ be $\mathbb{X}\times\mathbb{Y}\rightarrow\mathbb{R}$? Why is g’s domain $\mathbb{X}\times\mathbb{Y}$?
- Could the authors explain what PGA is?


**Limitations:**

The paper does not explicitly discuss the limitations and potential negative societal impact.

**Strengths And Weaknesses:**

Strengths:
- Addressing the challenges of large mini-batch training is important and timely.
- Technical aspects of the paper are reasonably clear.
- The proposed method is simple and effective.

Weaknesses:
- The paper heavily builds on the findings of existing works. It is hard to argue for technical novelty.
- I think the paper could be better organized. The current presentation has several separate ideas floating around lacking a clear central theme. Besides, I find the detailed discussion of previous work in the middle of the method section very distractive.
- The paper should include experiments on other pretrained transformers, as well as other datasets.

---

> ### Author Response · Authors · 2022-08-01
> **Response to Reviewer oBzn's comments**
>
> Thank you very much for taking the time to review our work and providing your useful feedback. Our response is listed below.
>
> ***\*Q1: The paper heavily builds on the findings of existing works. It is hard to argue for technical novelty.\****
>
> A1: Thank you. It is not our intention to propose a generic and new adversarial training algorithm. The main focus of this work is a careful evaluation of the effects of adversarial noise on computational efficiency and generalization in the NLP domain, varying the batch sizes. To the best knowledge, we are not aware of other work that has done such characterization. We also identify that large-batch adaptation with adversarial noise requires careful injection of lightweight adversarial noises to overcome the computation-vs-generalization dilemma and propose a recipe on how to do that. As you can see from our analysis in section 4.2, there are quite a few alternative choices, and our contribution here is to identify the most effective techniques to solve the problem.
>
> ***\*Q2: Concerns about having discussion of previous work in the middle of the method section.\****
>
> A2: Thanks a lot for pointing this out. We put the discussion of previous work such as "adversarial training preliminaries" in the middle of the method section to cater audience who may not be familiar with adversarial training. However, based on this feedback, we realize that this can be a distraction for an audience who may already be familiar with the content. We will move the preliminaries into the background section.
>
> ***\*Q3: Concerns about lacking a clear central theme.\****
>
> A3: Thanks for pointing out the confusion. We think this concern is raised because of two aspects: (1) The discussion of previous work in the design section somehow distracts the main theme a bit, as the reviewer pointed out in Q2. (2) Our main optimizations are tailored to address the specific challenges from PGA based on our analysis in Section 3.2 (as shown in Figure 2 and Figure 3). However, we did not give a high-level overview of those challenges and optimizations until Section 3.2, which might be a bit too late. On the former, we will address it as mentioned in A2. On the latter, we will add a short paragraph at the beginning of the design section to give a high-level flavor of our proposed method and the challenges it addresses. Please let us know if we interpret your comment correctly, and whether our proposed revision addresses your concern.
>
> ***\*Q4: Experiments on other pretrained transformers and datasets.\****
>
> A: We overall agree that evaluating more models and datasets helps strengthen the work. We evaluate our approach on BERT and RoBERTa-large using GLUE benchmark suites because they have been widely used as baselines by prior state-of-the-art studies such as [1-2] that we compare against in the paper. We focused primarily on encoder type of models because they are by now the most popular ones where downstream adaptation relies on upon, as can be viewed from public model repo such as https://huggingface.co/models. We also have a slightly more preference for evaluating the optimizations in depth. That's why you see a separate analysis section (Section 4.2) that focuses on ablation studies, curvature analysis, comparison with alternative noise distributions, alternative perturbation locations, scalability tests, generalization analysis with longer training time, and different batch sizes. We hope this in-depth analysis provides more insight to the readers on how adversarial noise affects the computation efficiency and generalization of model adaptation. We plan to extend the current work to more pretrained transformers, such as decoder type of models such as GPT style models, but that would require some additional careful analysis on how adversarial noise impacts generative models. We will add a discussion in the paper and  leave it as future work.

---

> > ### Comment · Reviewer_oBzn · 2022-08-03
> > **Thanks for the response**
> >
> > After reading other reviews and the authors' response, I decide to keep my rating unchanged.

---

> > > ### Author Response · Authors · 2022-08-03
> > > **Response to Reviewer oBzn**
> > >
> > > Thank you for the reply. We are sorry to hear that our response does not change your impression of the paper. Can you let us know which parts of our response help address your concerns and which parts are not? For parts that are not addressed. Can you kindly provide concrete suggestions on how we may improve? For example, for our proposed revision, do you think it helps address your concerns about the structure of the paper? What specific models/datasets are you of particular interests? This would help us understand your perspective better and also further improve the work.

---

> > > > ### Comment · Reviewer_oBzn · 2022-08-04
> > > > **Followup**
> > > >
> > > > I appreciate the authors' efforts in addressing my concerns. Those on presentation should be fairly easy to address.
> > > >
> > > > As confirmed in the response, the major contribution of this work can be characterized as an empirical evaluation of existing approaches in NLP applications. As such, I would expect that the paper covers a broader class of models/tasks to have a larger impact. I understand that this would take additional resources, and I respect the authors' decision to leave it to future work, but I cannot increase the rating in this case.

---

> > > > > ### Author Response · Authors · 2022-08-04
> > > > > **Response to the followup**
> > > > >
> > > > > Thank you. We understand. There are many different ways to advance the state-of-the-art. We choose to focus on the evaluation in depth rather than width. As such, we focus our studies on the curvature analysis, comparison with alternative noise distributions, alternative perturbation locations, the impact of the strength of adversarial noise on generalization and scalability, training with normal and large training budgets, the impact of adversarial noise on different batch sizes, and many other ablation studies in this paper.
> > > > >
> > > > > We would really appreciate it if you can provide concretely what pre-trained model or dataset in your mind that should be included.

---

> > > > > > ### Comment · Reviewer_oBzn · 2022-08-04
> > > > > > **Suggestion on additional experiments**
> > > > > >
> > > > > > Additional text encoding tasks with longer input would complement GLUE well. Maybe QA is worth looking into. I would also be curious to see how ScaLA performs on (conditional) generation with T5/BART and variants.
> > > > > >
> > > > > > I understand that conducting analysis with depth in these additional settings is infeasible given the limited turnaround time. If the authors could share some early findings and a clear roadmap for further investigation to include in the revision, I would be more than happy to revise my review.

---

> > > > > > > ### Author Response · Authors · 2022-08-05
> > > > > > > **Response to suggestions**
> > > > > > >
> > > > > > > Thank you for the concrete suggestions! Let us work on it and get back to you.

---

> > > > > > > ### Author Response · Authors · 2022-08-08
> > > > > > > **Reply to Reviewer oBzn (Part 1)**
> > > > > > >
> > > > > > > Thank you for the time and diligence that you put into this review. Here, we would like to share a status update on our investigation of ScaLA on the pre-trained T5-base and SQuAD-v2 dataset. In the end, we will also include a plan on what to include in the final revision.
> > > > > > >
> > > > > > > On our side, in the past few days, we extended ScaLA to support T5 and SQuAD-v2 dataset. We prioritized the implementation and experiments of this setup because we would like to analyze a case that covered both the encoder-decoder style model, QA task, and longer input lengths under the limited time constraint. Furthermore, since there have been very few studies on how adversarial training affects the generalization of generation tasks, we also feel this can be a good reference point for future studies. We used T5-base (220M) model from the HuggingFace model zoo’s checkpoint (https://huggingface.co/t5-base) and finetuned it on SQuAD-v2 with T5ForConditionalGeneration. We follow prior practices to set the sequence length of SQuAD-v2 to 384.
> > > > > > >
> > > > > > > To establish the baseline, we followed the hyperparameter setup suggested by HuggingFace (e.g., batch size 12, learning rate 3e-5) to fine-tune SQuAD-v2 and did an additional grid search of learning rates (1e-5, 3e-5, 5e-5, 7e-5, 9e-5, 1e-4, 3e-4) as we did in the manuscript for other datasets. Our reproduced baseline achieved better results than the ones reported in mrm8488/t5-base-finetuned-squadv2 · Hugging Face (EM/F1=78.53/81.71 vs. 77.64/81.32). The baseline took 2 hours and 10 minutes to finish one run on Nvidia V100 GPU.
> > > > > > >
> > > > > > > |                                  |     EM       |     F1       |
> > > > > > > |----------------------------------|--------------|--------------|
> > > > > > > |     Baseline                     |     77.64    |     81.32    |
> > > > > > > |     Our reproduced baseline    |     78.53    |     81.71    |
> > > > > > >
> > > > > > > We then applied ScaLA and did a preliminary analysis on how it impacts the adaptation speed and generalization of this task. In particular, we applied ScaLA to adapt pre-trained T5 to SQuAD-v2 with a batch size of 1k. We used the square root scaling rule to scale the learning rate as we increase the batch sizes. We did not tune other hyperparameters of ScaLA and used the same perturbation clipping radius and step size as in the manuscript (described in Appendix B) to generate adversarial noise. Overall, the observations on this task are in line with the observations we obtained from the pre-trained BERT/RoBERTa and the GLUE benchmarks in the manuscript:
> > > > > > >
> > > > > > > First, ScaLA achieves 0.37/0.84 points higher EM/F1 scores (78.98/82.55 vs. 78.53/81.71) with a 7.2x faster adaptation speed (2 hours 10 minutes vs. 18 minutes) than the baseline. This is achieved by (1) using large batch sizes to improve the aggregated training throughput, and (2) using lightweight injected adversarial noises to improve generalization.
> > > > > > >
> > > > > > > Second, compared with FreeLb, ScaLA achieves a similar EM/F1 score but with a 3.6x faster speed. ScaLA is faster because it removes redundant perturbation steps and leverages delayed perturbations, greatly reducing the overhead from adversarial noise.
> > > > > > >
> > > > > > > Third, the scalability of the baseline is limited by the small batch sizes it uses, causing severe under-utilization when scaling out the adaptation to multiple GPUs, e.g., it either cannot leverage multiple GPUs due to insufficient samples per batch (i.e., when the number of samples per batch is smaller than the number of GPUs) or does not have sufficiently large compute granularity to keep multiple GPUs running with aggregated high throughputs. On the other hand, although increasing the batch sizes and learning rates allows the adaptation to finish in a much shorter time (from 2 hours 10 minutes to 8 minutes), the final accuracy is lower than the baseline. In contrast, ScaLA allows the adaptation to match and even sometimes exceed the baseline accuracy while offering a much faster adaptation speed.
> > > > > > >
> > > > > > > |                 |     #GPUs    |     Batch   size    |     EM       |     F1       |     Training   time    |
> > > > > > > |-----------------|--------------|---------------------|--------------|--------------|------------------------|
> > > > > > > |     Baseline    |     1        |     12              |     78.53    |     81.71    |     2h10m              |
> > > > > > > |     Baseline    |     16       |     12              |     N/A      |     N/A      |     N/A                |
> > > > > > > |     Baseline    |     16       |     1024            |     78.27    |     81.47    |     8m                 |
> > > > > > > |     FreeLb      |     16       |     1024            |     79.04    |     82.65    |     66m                |
> > > > > > > |     ScaLA       |     16       |     1024            |     78.98    |     82.55    |     18m                |

---

> > > > > > > > ### Author Response · Authors · 2022-08-08
> > > > > > > > **Response to oBzn (Part 2)**
> > > > > > > >
> > > > > > > > We also conducted some ablation experiments to analyze the impact of different components in ScaLA for T5/SQuAD-v2. In short, we still see evidence that the number of perturbation steps and when to inject adversarial noise are crucial to both achieving fast adaptation speed and retaining high accuracy. Without PGA-1, the EM/F1 scores only increase marginally (e.g., <0.1), but the training time increases by 2.55x. This finding is consistent with the one in the manuscript: one-shot perturbation is sufficient to match the accuracy from the baseline without adding too much redundant computation. Without delayed perturbation, the adaptation increases by 1.44x. The EM/F1 scores increase by 0.22 and 0.14 points respectively, which are slightly higher than what we observed in the manuscript for GLUE (e.g., 0.1 points).
> > > > > > > >
> > > > > > > > |                                         |     EM       |     F1       |     Training time    |
> > > > > > > > |-----------------------------------------|--------------|--------------|------------------------|
> > > > > > > > |     ScaLA                               |     78.98    |     82.55    |     18m                |
> > > > > > > > |     ScaLA   w/o PGA-1 (e.g., PGA-3)     |     79.01    |     82.57    |     46m                |
> > > > > > > > |     ScaLA   w/o delayed perturbation    |     79.2     |     82.69    |     26m                |
> > > > > > > >
> > > > > > > > We hope these results provide additional data points to help understand the impact of adversarial noise on the computation efficiency and generalization of adapting pre-trained transformers for natural language understanding. For the next version of the paper, we plan to:
> > > > > > > >
> > > > > > > > 1. Fully flush out the above evaluations and add them to the paper.
> > > > > > > > 2. Increase the coverage of our evaluation results by adding additional results of BERT/RoBERTa on SQuAD.
> > > > > > > > 3. If time and compute resources are permitted, we will also add evaluation results on BART.
> > > > > > > >
> > > > > > > > Apart from these experiments, while we were working on T5/SQuAD, we think there might be additional experiments that could be interesting to perform:
> > > > > > > >
> > > > > > > > 4. Add additional experiments to test the scalability of ScaLA on larger scale T5 such as T5-large, a 770M parameter model, and presumably T5-3b (3 billion parameters), where the adaptation time becomes even more of a bottleneck.
> > > > > > > >
> > > > > > > > Please let us know your thought, and if any questions remain, we are happy to engage in further discussion!

---

> > > > > > > > > ### Comment · Reviewer_oBzn · 2022-08-08
> > > > > > > > > **Followup**
> > > > > > > > >
> > > > > > > > > Thanks for following up with the additional results and roadmap. I have updated my review accordingly.

---

> > > > > > > > > > ### Author Response · Authors · 2022-08-09
> > > > > > > > > > **Response to Reviewer oBzn**
> > > > > > > > > >
> > > > > > > > > > We are glad that your concerns are mostly addressed. This is a very positive NeurIPS experience for us, and we sincerely thank you for your constructive comments and active discussions that helped improve the paper.

---

### Official Review · Reviewer_eYN9 · 2022-07-10

**Rating:** 5
**Confidence:** 3
**Soundness:** 2 fair
**Presentation:** 3 good
**Contribution:** 3 good

**Summary:**

This paper proposes a ScaLA method to improve the computational efficiency of the fine-tuning of pre-trained language models while preserving generalization. ScaLA uses a large batch size to lower the total training iterations and employs adversarial training to mitigate the generalization drop brought by using a large batch size. This idea is quite counter-intuitive to me as it aims to lower the computational overhead, but introduces a more time-consuming adversarial training part. However, by using techniques such as one-shot adversarial examples and delayed perturbation injection, ScaLA is able to accelerate the adversarial training part and thus lower the overall computational overhead, which is empirically justified by experiments over a range of natural language understanding tasks.

**Questions:**

Kindly refer to my suggestion in the weakness part.

**Limitations:**

I do not see any potential negative social impact.

**Strengths And Weaknesses:**

Strengths:
1. The paper is well-written and the method is clearly presented.
2. The empirical result looks strong.

Weakness:
1. The proposed method looks like a naive combination of two existing techniques and thus to some extent incremental.
2. It would help if the authors can include a more in-depth analysis of why large batch size does harm to generalization and how this issue can be addressed (or even best addressed) by adversarial training. The current version focuses more on empirical validation.

===========After Rebuttal===========

Thank the authors for the response. After reading all the reviews and responses, I decided to keep my rating unchanged.

---

> ### Author Response · Authors · 2022-08-01
> **Response to Reviewer #eYN9's comments**
>
> Thank you very much for taking the time to review our work and providing your useful feedback. Our response is listed below.
>
> ***\*Q1: Concerns regarding novelty.\****
>
> A1: We agree that each individual technique may not appear very new. Our main contribution here is on a careful evaluation of the effects of adversarial noise on computational efficiency and generalization in the NLP domain, varying the batch sizes. To the best knowledge, we are not aware of other work that has done such characterization. We also identify that large-batch adaptation with adversarial noise requires careful injection of lightweight adversarial noises to overcome the computation-vs-generalization dilemma and propose a recipe on how to do that. Our recipe is a result of the investigation, and it is not our intention to propose a generic and new adversarial training algorithm. Finally, our paper provides a theoretical analysis for training with adversarial noise in the context of large batch learning, which to our best knowledge, is novel.
>
> ***\*Q2: In-depth analysis of why large-batch size harm generalization.\****
>
> A2: Thank you for the suggestion. We agree. We provided an analysis of the eigenvalue of the model's Hessian to measure the steepness of loss landscape in three cases: (1) small batch size, (2) large batch size, and (3) large batch size with adversarial noise. Our finding is that large-batch adaptation often leads to large eigenvalues, indicating that the model trained with large-batch is more prone to ill-conditioning and less robust to small perturbation, a phenomenon that often correlates to poor generalization. In contrast, adversarial noise helps reduce the eigenvalues, inducing smoothness to the model and improving generalization. Please find the detailed analysis in the "curvature analysis" under Section 3.2. Finally, we agree that an in-depth theoretical analysis of why adversarial noise improves generalization is also very important. However, we expect it to be difficult since all state-of-the-art adversarial training methods lack such analysis. This is an open question, and we will leave it as future work.

---

### Official Review · Reviewer_tfV1 · 2022-07-12

**Rating:** 7
**Confidence:** 3
**Soundness:** 4 excellent
**Presentation:** 2 fair
**Contribution:** 3 good

**Summary:**

This paper addresses the problem of speeding up adaptation training by introducing an adversarial training algorithm that is compatible with large batch training. Previously, while being able to significantly speed up adaptation training, large batch training has been shown to harm the model's ability to generalize. Adversarial training algorithm helps, but the inner optimization steps brings a major computation overhead over other optimization algorithms. Motivated by these insights, this paper proposes ScaLA, which (1) sticks to the adversarial solution but reduces gradient computation steps to 1, and (2) introduces a delay in perturbation injection, in order to bring down the computation overhead. Experiments show that the proposed training regime is faster than previously-proposed solutions like FreeLb and LAMB, while also improving the performance of the model, which indicates stronger generalization capabilities.

**Questions:**

Please respond to the points made in the weaknesses & detailed comments part above.

**Limitations:**

Adequately addressed to the best of my knowledge.

**Strengths And Weaknesses:**

Strengths:

- The research problem is well-motivated by the related work survey and the preliminary analyses.
- The presentation of the solution is mostly clear, with the benefit of each part being clearly stated.
- Experiments show clear benefit over existing methods in terms of adaptation training speed, while also showing strong performance.

Weaknesses:

- It is unclear to me if any of the modification here is specifically targeted to speed-up the adaptation training process (as opposed to pre-training).
- There are a few non-ideal presentation choices, see below.

Some more detailed comments:

- Eqn. 2: the choice of using $x$ and $y$ as model parameters and perturbed embeddings is a very confusing one, especially since they were used as instances from the training data in (1).
- L130: It would be nice to introduce what exactly with and without communication means, or provide some citations for it. I have trouble understanding how would the non-communication case work.
- Algorithm 1 L9: I don't understand $\prod_{\omega}$ means here.

---

> ### Author Response · Authors · 2022-08-01
> **Response to Reviewer tfV1's comments**
>
> Thank you very much for taking the time to review our work and providing useful feedback. Our response is listed below.
>
> ***\*Q1: It is unclear to me if any of the modification here is specifically targeted to speed-up the adaptation training process (as opposed to pre-training.\****
>
> A1: Our approach should be applicable to pre-training, but we are careful not to claim scenarios that we have not thoroughly evaluated. Our current study primarily focuses on the adaptation phase, because there is a motivation to have fast adaptation and it provides a more controlled environment where the training objective aligns with the evaluation objective. In contrast, drawing conclusions via pre-training tasks is more complicated because: (1) the language model objective during the pre-training (e.g., MLM) may not correlate well with the downstream task's objective (e.g., classification), and (2) it would require much more time and resource to conduct a thorough investigation of the computation-vs-generalization trade-offs from adversarial noise for pretraining tasks because pre-training often happens on a much larger open-domain data. We agree that pre-training is an important and interesting task to look at. We would like to discuss it in the paper and explore it as future work.
>
> ***\*Q2: Eqn. 2: the choice of using x and y as model parameters and perturbed embeddings is a very confusing one, especially since they were used as instances from the training data in (1).\****
>
> A2: Thanks for pointing it out. We originally used $x$ and $y$ in Eqn.(2) because they add some simplicity when performing theoretical analysis. However, with this comment, we realized that we slightly abused the notation such that x,y caused a conflict with the training data notation Eqn.(1). We see two ways to fix it. The first is to update Eqn. (1) such that its notations are consistent with Eqn. (2). This would require minimal change. The second option is to change the notation in Eqn. (2) by replacing $x$ with $\theta$ to represent the model parameters and y with $\tilde{\xi}$ to represent the perturbed embeddings. This would also resolve the conflict, and we will also update x and y in the algorithm and theoretical analysis/proofs to make them consistent across the manuscript. Our current plan is to take the first option for the revision unless you have a different preference. Also, please let us know if this reply helps resolve the confusion. We are happy to provide further clarification.
>
> ***\*Q3:  It would be nice to introduce what exactly with and without communication means, or provide some citations for it. I have trouble understanding how would the non-communication case work.\****
>
> A3: Thank you for pointing this out. No-communication is a configuration we created to help analyze the communication overhead in distributed training. It replaces the AllReduce communication primitive with a no-op implementation. Such a config will not generate the same convergence results as the baseline config with AllReduce communication enabled because no gradients are communicated and averaged among parallel workers. However, it allows us to analyze the fraction of training time spent doing communication and identify potential communication bottlenecks. We will add an explanation to the manuscript.
>
> ***\*Q4:  Algorithm 1 L9: I don't understand $\Pi_\omega$ means here.\****
>
> A4: Thank you for pointing out the confusion. Generating adversarial noise requires solving an extra maximization problem, which is often solved with Projected Gradient Ascent (PGA) that projects the result of the gradient step with a constraint. In our case, we project the result of the gradient ascent update into an $\ell_\infty$ ball of diameter $\omega$ around the original input embeddings, which we denote as $\Pi_\omega$. We will add the clarification in the manuscript.

---

### Meta-Review · Area_Chair_osuJ · 2022-08-21

**Recommendation:** Reject
**Confidence:** Less certain

**Metareview:**

This paper presents ScaLA, a method to improve the efficiency of finetuning of a pretrained language model using adversarial training. Experiments confirm that ScaLA allows faster finetuning compared to standard approaches without reducing accuracy.

I think this is a nice approach that can have a lot of impact for those who would like to finetune large language models. However, the reviewers have concerns regarding novelty of the paper and would like to see the proposed method to be tested for other larger scale pretrained models as well as more analysis. I encourage the authors to incorporate these changes and resubmit to another conference.

**Award:**

No

---

### Decision · Program_Chairs · 2022-09-14

Reject